Corrected: Author correction

# A mixed antagonistic/synergistic miRNA repression model enables accurate predictions of multi-input miRNA sensor activity

Jeremy J. Gam[1,2], Jonathan Babb[1,2] & Ron Weiss[1,2]

MicroRNAs (miRNAs) regulate a majority of protein-coding genes, affecting nearly all biological pathways. However, the quantitative dimensions of miRNA-based regulation are not fully understood. In particular, the implications of miRNA target site location, composition rules for multiple target sites, and cooperativity limits for genes regulated by many miRNAs have not been quantitatively characterized. We explore these aspects of miRNA biology at a quantitative single-cell level using a library of 620 miRNA sensors and reporters that are regulated by many miRNA target sites at different positions. Interestingly, we find that miRNA target site sets within the same untranslated region exhibit combined miRNA activity described by an antagonistic relationship while those in separate untranslated regions show synergy. The resulting antagonistic/synergistic computational model enables the high-fidelity prediction of miRNA sensor activity for sensors containing many miRNA targets. These findings may help to accelerate the development of sophisticated sensors for clinical and research applications.

[1] Department of Biological Engineering, Massachusetts Institute of Technology (MIT), Cambridge, MA 02139, USA. [2] Synthetic Biology Center, MIT, Cambridge, MA 02139, USA. Correspondence and requests for materials should be addressed to R.W. (email: rweiss@mit.edu)

MicroRNA (miRNA) molecules are short sequences of non-coding RNA that are important for post-transcriptional regulation of mRNA. Despite active study of miRNAs since their discovery, several aspects of miRNA repression remain unknown or controversial[1]. For instance, many of the proteins and mechanisms involved in miRNA repression and interactions between them have yet to be elucidated[2,3]. Also, most studies have focused on miRNA target sites in the 3′ UTR[4], but recent research has shown that targets in the coding sequence and 5′ UTR can be important for modulating activity, especially in combination with other target sites[5–7]. Due to incomplete study of these interactions between target sites, there has been a lack of consensus for the importance of target sites outside of the 3′ UTR and also insufficient knowledge to generate design rules and models for transcripts regulated by many miRNAs simultaneously. We anticipate that the ability to explain and predict effects of simultaneous repression by multiple miRNAs will become increasingly important for understanding miRNA regulation, since nature is replete with examples of highly miRNA-regulated genes. On average 7.3 different miRNAs repress each miRNA-regulated gene and 47 distinct genes are regulated by >40 miRNAs[8], with p21Cip1/Waf1 experimentally verified to be targeted by 28 miRNAs[9]. Additionally, emerging evidence indicates a class of transcripts regulated by simultaneous 5′ and 3′ UTR targets of the same miRNA[6].

The ability to predict multi-miRNA repression may also be applied to create better nucleic acid-based therapeutics (e.g., ones that are regulated dynamically by complex biomarker profiles). We are especially interested in using miRNAs as indicators of cell type and cell state, since there are thousands of distinct miRNAs which regulate >5300 genes across almost all cellular pathways[10–12]. Several studies have used miRNA profiles to identify diseases including cancer[13], Alzheimer's disease[14], and heart disease[15], while we and others have shown that genetically encoded miRNA sensors can be constructed by placing miRNA target sites in the UTRs of a reporter[16–19]. These genetically encoded miRNA sensors (which sense a single miRNA input) and cell classifiers (which sense multiple miRNA inputs simultaneously) can provide information about disease state, actuate responses in cells specifically expressing either a diseased or healthy miRNA profile[16–18], distinguish between subtypes of cells in vivo[19], and help biologists study complex processes like stem cell differentiation[20]. While most efforts have focused on sensors measuring a single miRNA at a time, multi-input miRNA classifiers more closely mimic endogenous biological regulation in that many miRNAs (comprising a miRNA profile) can regulate a single transcript, improving specificity and redundancy.

To improve our ability to predict regulation from multiple miRNAs, we created a large library of reporter constructs with composable miRNA target sites and used them in various combinations to explore the effects of multi-miRNA regulation from 5′ and 3′ targets. We use highly expressed synthetic miRNA sensors and modeling to probe the limits of miRNA regulation, since quantitative measurements made at biological extremes can provide mechanistic insight otherwise difficult to obtain via conventional knockout or sequencing based techniques[1,21]. We found that miRNA target site interactions follow an antagonistic/synergistic (Ant/Syn) model where sets of miRNA target sites exhibit antagonistic interactions within the same UTR (i.e., the amount of knockdown depends strictly on the miRNA target sites with highest activity), and synergistic interactions across UTRs (i.e., knockdown is a multiplicative combination of miRNA target sites). In contrast to previous computational models[22,23], our Ant/Syn model accurately predicts simultaneous repression effects from many different miRNAs.

The desire for sophisticated miRNA classifier designs that perform increasingly more complex operations necessitates a deeper understanding of the composition rules that govern regulation of transcripts by many miRNAs. In this study we introduce a workflow for measuring output of single-input miRNA sensors in cell lines, characterizing miRNA activity from miRNA sensor data using a biochemical model, utilizing the measured miRNA activity to make accurate predictions of multi-input miRNA classifiers using the Ant/Syn model, and testing the best classifier candidates in cells. The accurate predictions generated by the model reduce or eliminate the need for multiple iterations of physical classifier designs by instead simulating a large range of classifier designs in silico and testing only the best candidates experimentally. Interestingly, the use of our Ant/Syn model often leads to improved classifier designs that are counterintuitive. For instance, classifiers may perform markedly better with rearranged target sites or even with fewer target sites, depending on the application. These and other design rules, both intuitive and counterintuitive, can help explain aspects of the regulation of endogenous transcripts and improve the design of miRNA sensor circuits, illustrating the power of convergence between quantitative biological modeling and analytical synthetic circuit design.

## Results

**miRNA activity characterization by modeling of miRNA sensors.** We designed, constructed, and sequence validated a miRNA sensor library containing all 620 sequences of mature human miRNAs designated as high confidence in miRBase 21[24]. Our library enables high-information-content screening of miRNA activity in cells and also serves as a source for sequence-validated templates of miRNA targets when building multi-input sensors. We synthesized miRNA target site sets bearing four repeats of the sequence perfectly complementary to the miRNA and inserted them into the 3′ UTR of a reporter construct (Fig. 1a). Throughout this work, a miRNA target site "set" refers to four repeats of a given miRNA target site. We chose to include four target sites per set since minimal increase in dynamic range was obtained with >4 targets (Supplementary Figs. 1 and 2) and used perfectly complementary target sites to reduce the possibility of miRNA sponging effects[25] while increasing the dynamic range of repression. The resulting single-input sensors demonstrate up to several-hundred-fold dynamic range, indicating the utility of miRNAs as potent biomarkers (Fig. 1b, c). To our knowledge, ours is the largest miRNA sensor library reported to date (the previous largest containing target sites for 291 miRNAs[26]) and the only one allowing for Golden Gate assembly[27] from single-input sensors into multi-input classifiers in a single assembly step. The library will be available on Addgene for researchers to use in advancing the understanding of miRNA biology and the development of better DNA and RNA-based therapies.

We implemented a computational miRNA repression model to better understand and quantify the behavior of our sensors and classifiers (Supplementary Note 1). The model for single-input sensors is based on previous deterministic models[22,23] for miRNA activity, which we extended to multi-input sensors using the Ant/Syn model. At the single-input level, the model comprises the following reactions: (1) mRNA and protein molecules for two fluorescent reporters are transcribed and translated at first order rates and are degraded, (2) mKate2 reporter mRNA may be reversibly bound by a miRNA-containing complex which may then catalytically degrade the bound mRNA, (3) parameters corresponding to miRNA concentration ($M$, the total effective number miRNA molecules either free or bound to reporter mRNA), and Michaelis constant ($K_m = \langle k_{off} + k_{cat}\rangle/k_{on}$, a measure related to the repression strength per miRNA molecule) are

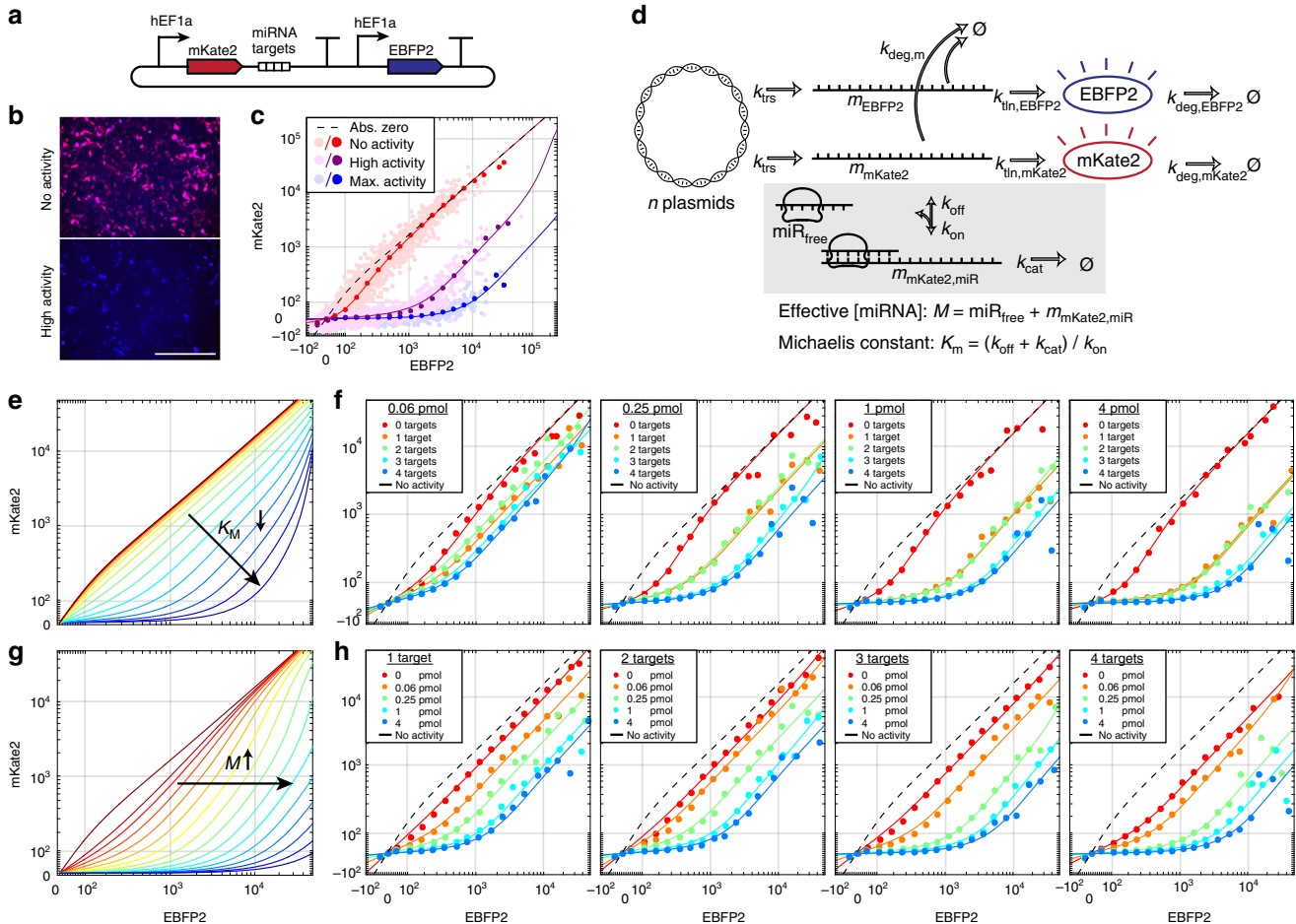

**Fig. 1** A biochemical model explains miRNA repression measured using miRNA sensors. **a** A schematic of the single-input miRNA low sensors used in this study. **b** Fluorescence microscopy in HEK293FT cells for a control sensor with miRNA target sites with no activity (FF4) compared to a sensor with target sites for a high activity miRNA (miR-106a-3p). Both images are merged blue/red channels and the scale bar indicates 400 μm. **c** Corresponding flow cytometry data in HEK293FT cells for no activity (red) and high activity (purple) sensors. Data for a positive control using exogenous siRNA to repress mKate2 is shown for reference (blue). Light scatter points correspond to data; dark points represent median values for data binned by EBFP2 fluorescence; lines indicate fits to the data using a biochemical model; dashed line shows modeling results with absolutely zero miRNA activity. Fluorescence is shown in terms of arbitrary units and with logicle scaling for the axes[46]. **d** The single-input repression model annotated with relevant species and rate constants. Each cell is transfected with some number of plasmids ($N$) which are transcribed (at rate $k_{trs}$) and translated ($k_{tln,EBFP2}$ and $k_{tln,mKate2}$) to yield fluorescent proteins. mKate2 transcripts are reversibly bound by a miRNA-containing complex ($miR_{free}$) forming a bound species ($m_{mKate2,miR}$) which is catalytically degraded at some rate ($k_{cat}$). Two parameters, effective miRNA concentration ($M$) and Michaelis constant ($K_m$) fully characterize miRNA activity. RNA and protein species are non-specifically degraded at some rate. Fixed rate constants are approximated from literature values (Supplementary Table 3). **e** Model predictions for decreasing $K_m$ while holding $M$ constant. As $K_m$ decreases, repression increases while switching threshold is maintained. **f** Experimental perturbation of $K_m$. HEK293FT cells, which exhibit low endogenous miR-21-5p activity, were transfected with sensors containing varying numbers of target sites for miR-21-5p and varying amounts of exogenous miR-21-5p mimic. As the number of target sites was increased (decreasing $K_m$), repression increased as predicted. **g** Model predictions for increasing $M$ while holding $K_m$ constant. As $M$ increases, the switching threshold increases and repression increases as a result. **h** Experimental perturbation of $M$. As miR-21-5p mimic concentration is increased, repression increased as predicted

fit using fluorescence data obtained experimentally from our sensors and together accurately describe the behavior of a given miRNA sensor (Fig. 1d). In concordance with previous models[22,23], our model predicts thresholding behavior with three output regimes when sensing a single miRNA. At low EBFP2 expression (i.e., low transfection efficiency), a repressed regime exists where mKate2 is significantly repressed relative to EBFP2. At high EBFP2 expression, an excess of reporter mRNA is present which saturates the miRNA machinery, resulting in a derepressed regime. And at intermediate EBFP2 expression, a threshold regime switches between repressed and derepressed behavior.

We tested the ability of our single-input repression model to explain changes to experimental perturbations of $M$ and $K_m$. For decreasing $K_m$, our model predicts no shift in threshold region—

only a decrease of output within the repressed region (Fig. 1e). We modulated $K_m$ experimentally by varying the number of target sites for miR-21-5p in the sensor. As expected from the model, we observed that increasing the number of target sites increased $k_{on}$, resulting in a decrease in $K_m$ and increase in repression (Fig. 1f). This result was also confirmed for endogenous high activity miRNA in HEK293FT and HeLa cells (Supplementary Fig. 1). For increasing total miRNA concentration ($M$), the model predicts a shift in threshold region to the right and a concomitant decrease of output within the repressed region (Fig. 1g). We modulated $M$ by transfecting various amounts of miR-21-5p mimic in HEK293FT cells which have low endogenous miR-21-5p activity. The expected decrease in output was observed across increasing miR-21-5p amounts (Fig. 1h).

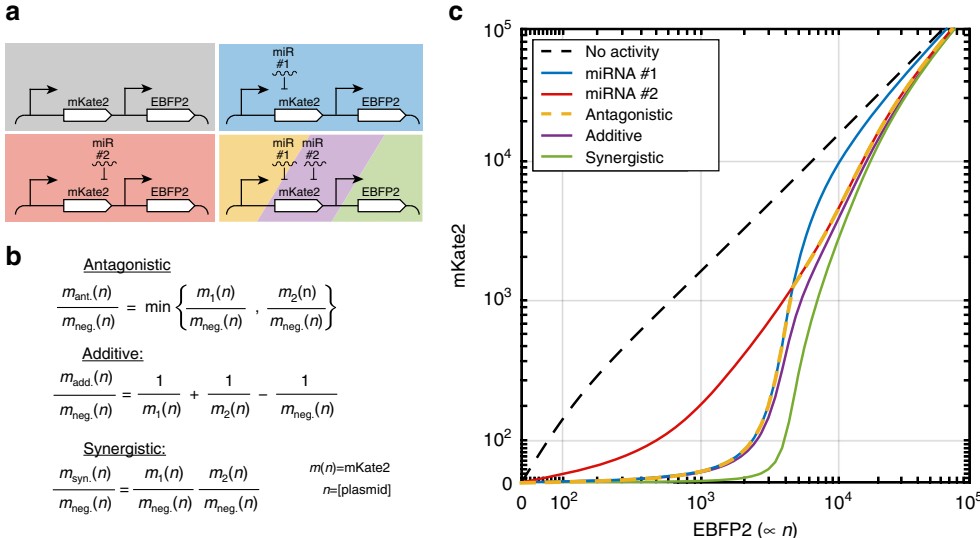

**Fig. 2** Methods for combining miRNA activities in predictions of 2-input low sensors. **a** Schematics for 0-input control (gray), 1-input (blue and red), and 2-input miRNA sensors (yellow/purple/green stripe). The 1-input miRNA sensors are first characterized, and the characterization used to predict the behavior of multi-input sensors. **b** Equations based on the Chou–Talalay method adapted here for predicting miRNA activity of 2-input sensors according to antagonistic, additive, and synergistic interactions. Predictions of free mKate2 mRNA concentration for the three interaction types ($m_{ant.}(n)$, $m_{add.}(n)$, and $m_{syn.}(n)$) are computed based on the characterized output of miRNA 1 ($m_1(n)$) and miRNA 2 ($m_2(n)$). All mKate2 mRNA concentration predictions are functions of the number of plasmids ($n$) in the cell, which is proportional to EBFP2 fluorescence. $m_{neg.}(n)$ refers to the output of the negative control where no miRNA target sites are present. General equations for cases with more than two miRNA inputs are presented in the Supplementary Note 1. **c** Calculation of 2-input predictions from single-input activity. Colors for the curves correspond with those from **a**. The repression curve for the negative control construct without target sites (black) is shown for reference. miRNA repression curves are illustrated for different single miRNAs (blue and red), which are generated by fitting $M$ and $K_m$ to sensor data using the single-input model illustrated in Fig. 1. The blue and red single-input repression curves intersect because in this example miRNA #1 exhibits lower $M$ and $K_m$ but higher $M/K_m$ compared to miRNA #2, leading to greater repression at lower EBFP2 expression levels and vice versa. Antagonistic predictions (yellow) closely follow the miRNA with highest activity (minimal mKate2 expression) for a given level of EBFP2 expression, additive predictions (purple) are similar but with increased overall activity when the two constituent miRNAs have similar individual activity, and synergistic predictions (green) always exhibit higher miRNA activity than either antagonistic or additive predictions. Generally, discrepancy between the two types of predictions varies depending on the number and miRNA activities of the single-inputs, with greater discrepancies when many similar activity single-inputs are combined

To experimentally determine the values of $M$ and $K_m$ for each of our miRNA sensors, we used a reverse transfection protocol[28] to introduce our library of sensor plasmids into common cell lines HEK293FT and HeLa. We then used computational methods to generate fits of $M$ and $K_m$ for each miRNA sensor in both cell lines, generating a list of miRNA activities. Fits based on the model captured the effects of miRNA-mediated repression well as indicated by low and normally distributed errors (Supplementary Fig. 3). Most miRNA sensors exhibited low or no miRNA activity (396 of 620 sensors or 64%), consistent with previous reports of miRNA sensor measurements in several cell lines (Supplementary Fig. 4)[26]. To compare miRNA activity to miRNA expression in our cell lines, we submitted HEK293FT and HeLa for small RNA sequencing and estimated miRNA expression levels using existing computational tools[29]. miRNA activity was weakly correlated with miRNA expression (Supplementary Fig. 4), again consistent with observations in other cell lines[26]. We also show that miRNA activities measured using the sensor library are reproducible, with two biological replicates showing good correlation of a $M/K_m$ metric which describes the maximal fold repression of a sensor due to miRNA activity (Supplementary Fig. 4). Analysis of miRNA activities in HEK293FT, HeLa, and HepG2 helped reveal miRNAs specific to each cell line relative to the others, which we later used to construct a HEK293FT cell classifier (Supplementary Table 1).

## Synergistic and antagonistic miRNA target set interactions.
Following the characterization of single-input miRNA sensors, we

sought to use the obtained parameters to make predictions for more complicated multi-input cell classifiers. Because our initial purely synergistic model did not provide accurate predictions for multi-input sensor function, we evaluated models that include additive, antagonistic, or synergistic interactions (Fig. 2). These three types of interactions were implemented as in the Chou–Talalay method[30], which comprises a standard set of equations used to determine whether inhibitors interact with each other. Additive predictions were made assuming mutually exclusive inhibitors, antagonistic predictions were made using similar assumptions but with a Hill coefficient value near zero, and synergistic predictions were made by multiplying contributions from each target site set (Fig. 2b, c and Supplementary Note 1). Here additive interactions indicate that combined repression from multiple inhibitors is equal to the 'sum' of their separate effects. For example, with two miRNAs giving 50% knockdown at a given plasmid concentration, additive interaction would give a final knockdown of 67%. Note that final knockdown is not 75% since that would indicate independent contributions to knockdown, which is a synergistic interaction. By definition synergistic and antagonistic interactions mark a deviation from the additive case. For the Ant/Syn model and taking the previous example of two miRNAs with 50% knockdown, independent (i.e., synergistic) contributions result in the intuitive 75% knockdown, while for the antagonistic case knockdown would be 50% since only the highest activity miRNA contributes to repression.

We then tested 3-input sensors (Fig. 3a) encoding several combinations of representative low and high activity miRNA

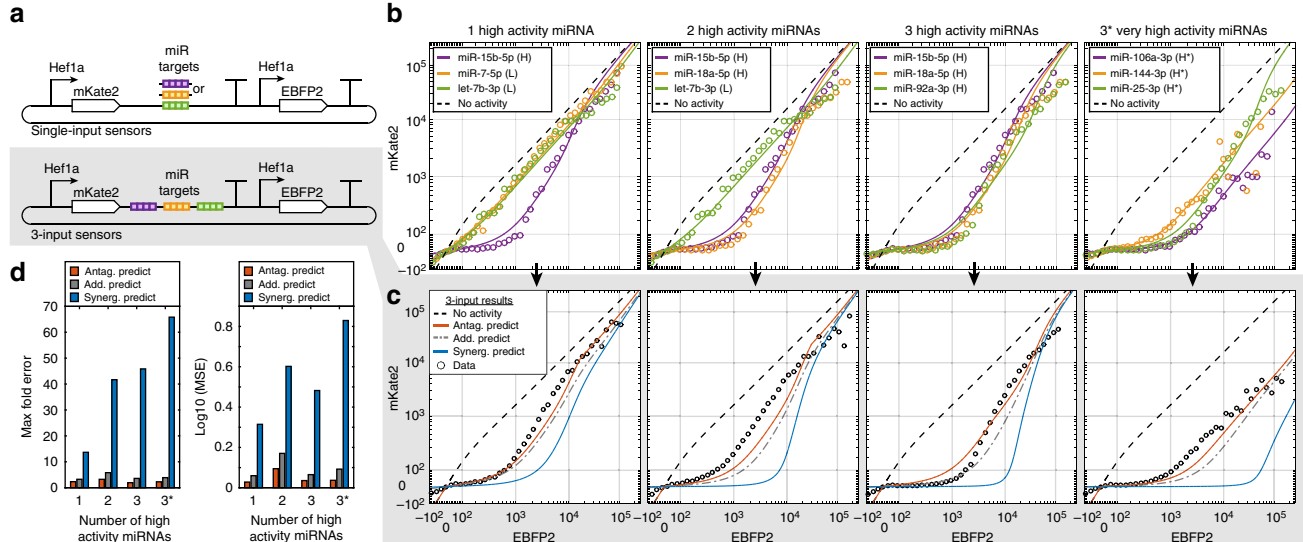

**Fig. 3** Interactions between miRNA target site sets in the 3′ UTR sensors appear antagonistic. **a** Schematic for single-input sensors bearing one set of target sites (top) and 3-input sensors bearing three different sets of target sites (bottom) where all miRNA target sites are located in the 3′ UTR. **b** miRNA repression data and model fits for different single-input sensors. Plots are ordered by increasing number of high activity miRNA target site sets. miRNA activities are denoted as low (L), high (H), or very high (H*). **c** Comparison of predicted miRNA repression to data obtained for 3-input sensors. Predictions for 3-input sensors are computed based on $M$ and $K_m$ parameters measured from single-input sensors. Using the assumption that miRNA target sites act antagonistically yields predictions (red) which are equivalent to taking the maximum activity (minimum mKate2 expression) of the three single-input sensors for each EBFP2 expression level. Additive predictions (gray dashed) are made using the Chou–Talalay method. Synergistic predictions are made by multiplying fold repression from each of the single-input sensors within each EBFP2 transfection bin. Predictions using the antagonistic or additive models are markedly better than those from a synergistic model, indicating that miRNA repression for multi-input sensors is not a simple multiplicative effect. While antagonistic and additive models were close in these examples, predictions can diverge drastically when several miRNA inputs are combined and when input activities are very similar. **d** Analysis of prediction errors for 3-input sensors. Errors are measured by computing the maximum fold difference between predictions and data across all bins of EBFP2 expression (max fold error) or by computing the mean squared error (MSE). For both metrics and for all tested combinations of low/high/very high (*) activity miRNAs, antagonistic interaction explains the data better than an additive or synergistic interaction

target site sets in the 3′ UTR of the output and compared obtained data to the predictions (Fig. 3b, c). Antagonistic interactions best predicted the multi-input data as indicated by lower maximal fold error and mean squared error, while additive and synergistic predictions consistently overestimated miRNA activity (Fig. 3d). Interestingly, inclusion of up to three high activity target sets (12 target sites) had minimal effect on increasing repression further than the single highest activity target set, i.e., the 3-input sensor exhibits activity reflective of only the highest activity target set.

To fully explore the general composition rules for miRNA target sets, we tried spacing target sets further apart or placed them in separate UTRs and investigated whether interactions were antagonistic, additive, or synergistic. We first tested separating target sets withing the 3′ UTR with spacers up to 600 bp but still observed antagonistic interactions even with the longest spacers (Supplementary Fig. 2). In contrast, we found that miRNA target sets in the 5′ UTR exhibited synergistic miRNA repression when combined with target sets in the 3′ UTR (Fig. 4). Based on these results we hypothesized that models for predicting miRNA activity would need to consider miRNA target set position in addition to $M$ and $K_m$.

**Ant/Syn model predicts multi-input miRNA classifier output.** Next we tested whether we could generate more accurate predictions than previous repression models by taking into account antagonistic interactions of miRNA target sets within UTRs and synergistic interactions across UTRs. This model, which we term an Ant/Syn model, first calculates (on a per transfection marker basis) the antagonistic interactions within each of the UTRs

(approximated as the minimum reporter expression observed for any single-input sensor within one UTR) and then calculates the synergistic interactions between the UTRs by multiplying the contributions of each UTR to obtain the final output (Supplementary Fig. 5). This is in contrast to the antagonistic-only model where only maximal repression for any single miRNA is taken, or the synergistic-only model where all contributions for each miRNA target site are multiplied, regardless of UTR position.

To test the Ant/Syn model experimentally, we built several variants of 4-input miRNA sensors (i.e., classifiers) bearing two miRNA target site sets in the 5′ UTR and two miRNA target sets in the 3′ UTR (Fig. 5a). We selected twelve miRNA target sites for study based on their miRNA activity measured from single-input sensors—four each of high, medium, and low activity miRNAs. We assembled and assayed reporter expression of 36 different 4-input sensors in HEK293FT cells representing a diverse panel of activities (Supplementary Table 2), one example of which is shown in Fig. 5b. Across 36 tested cases, the Ant/Syn model predicted final output similarly or better than the antagonistic-only or synergistic-only models (Fig. 5c). For all classifiers with behavior accurately predicted by the synergistic-only model, the antagonistic-only model underestimated activity, while classifiers with behavior accurately predicted by the antagonistic-only model had activity overestimated by the synergistic model.

**Ant/Syn-informed classifiers better distinguish cell types.** We next investigated whether the Ant/Syn model could be used to design better single-input sensors and multi-input cell classifiers containing miRNA targets in both 5′ and 3′ UTRs which would allow sensors and cell classifiers to leverage antagonistic and

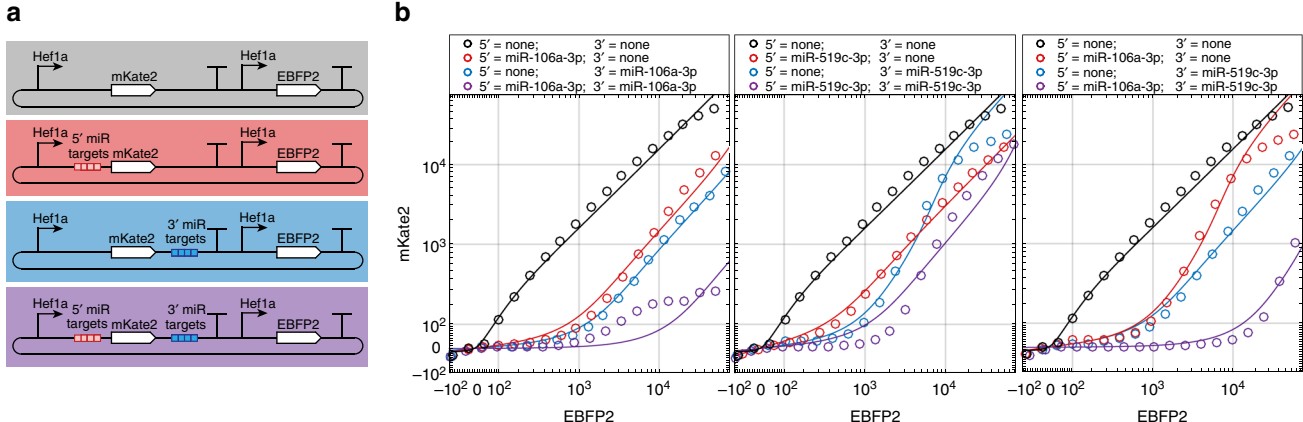

**Fig. 4** Positioning miRNA target site sets in separate UTRs yields synergistic interactions. **a** Diagram of tested constructs bearing target sites in 5′ UTR only (red), 3′ UTR only (blue), both 5′ and 3′ UTRs (purple), and no target sites (gray; shown as black in **b**). Constructs were designed to sense endogenous miRNAs with detectable activity in the tested cell line (HEK293FT). **b** miRNA repression data (circles) and model fits (lines) obtained for each of the four constructs with miR-106a-3p target sites, miR-519c-3p target sites, or a combination of the two. Constructs bearing target sites in both 5′ and 3′ UTRs showed greater repression than constructs with target sites in only a single UTR, demonstrating a synergistic effect not observed in constructs where target site sets were confined to the 3′ UTR only

synergistic interactions to obtain better sensitivity or specificity in classifying different cell types. Such multi-input constructs also inform principles by which endogenous miRNAs could utilize 5′ UTR target sites to enhance conventional activity from the 3′ UTR[6]. We built two versions of a sensor containing multiple target sets for miR-21-5p, which shows high activity in HeLa and HepG2 cells but low activity in HEK293FT cells. In one variant, target site sets were placed in only the 3′ UTR, while in another variant, sets were placed in both the 5′ and 3′ UTRs. We also constructed a 3-input cell classifier containing target sites for miR-21-5p, miR-23a-3p, miR-106b-3p where targets were either all in the 3′ UTR or separated into different UTRs (Fig. 6a).

When separately transfected into HEK293FT, HeLa, and HepG2 cells, all sensors and classifiers showed reporter knockdown in HeLa and HepG2 cells and retained high reporter expression in HEK293FT cells as expected (Fig. 6b). The classifiers with targets in separate UTRs showed a marked reduction in reporter expression for off-target cells at high transfection levels resulting in greater on/off ratio (280-fold on/ off for miR-21-5p, 210-fold for 3-input), in contrast to classifiers with targets only in the 3′ UTR (14-fold on/off for miR-21-5p, 29-fold for 3-input). Thus our new constructs with target sets in both UTRs achieved ~20-fold (miR-21-5p sensor) and 7-fold (3-input classifier) improved on/off ratio, compared to constructs with target sets in only the 3′ UTR. The reduction in false positive output is consistent with the synergistic interactions anticipated for miRNA target sets placed in separate UTRs. We then tested whether we could enhance classification in a mixture of cell types. We chose to test in a co-culture of HEK293FT, HeLa, and HepG2 cells where HEK293FT cells expressed genomically integrated EYFP in order to provide an independent measure of whether cells classified as HEK293FT by our classifiers were indeed HEK293FT in origin (Supplementary Fig. 6). Reporter expression in co-culture was similar to that in separate transfections, with classifiers designed to utilize synergistic interactions exhibiting reduced false positive rate at high transfection levels (Fig. 6c). Quantification of sensitivity and specificity showed that while sensitivity was high (>95%) for all designs, specificity was improved (>85%) only in the new designs based on the Ant/Syn model (Fig. 6c). We chose to place only miR-21-5p target sets into the 5′ UTR of these classifiers to minimize the length of

the 5′ UTR—reducing the potential for 5′ UTR structure effects on reporter expression—while still utilizing synergism across UTRs.

While it is possible to duplicate all miRNA target sites across both UTRs, this may not be desirable in cases where there is a size constraint, where target sites bear an 'ATG' (which can initiate translation of an upstream ORF), or when it is important to have synergistic interactions for different miRNA species (with duplicated target sets only the single miRNA species with highest activity would show synergism). Moreover, target site duplication can be adjusted to optimize the trade-off between sensitivity and specificity according to whichever is more important for the given application. For example, high sensitivity may be more important for cell classifiers designed to specifically kill cancer cells to reduce the chance of cancer cells escaping, while high specificity may be more important for classifiers designed to induce differentiation of a particular cell type at a particular differentiation stage (e.g., driving liver differentiation in a subset of cells in the endoderm but absolutely not elsewhere). We tested several variants of cell classifiers with and without target site duplication and examined the effects of duplication on classifier sensitivity and specificity. The Ant/Syn model predicts that high sensitivity and lower specificity will be obtained by removing target set duplication from both UTRs, which we observed in the data (Fig. 7). The effects of miRNA target location and duplication were further analyzed using receiver operator characteristic (ROC) curves (Fig. 8). Again, classifiers with targets in both UTRs demonstrated improved accuracy and area under the curve (AUC) compared to those with targets in only the 3′ UTR (Fig. 8a, b), while those with target set duplication showed trade-offs between specificity and sensitivity (Fig. 8c) and in certain cases improved performance without target duplication (Fig. 8d).

## Discussion

We report a framework for making accurate predictions of multi-input miRNA sensors using parts level characterization. Several aspects of miRNA target regulation comprise our current design framework. First, concatenation of miRNA target sites in a UTR increases repression of a target reporter up to approximately 4 repeats (Supplementary Figs. 1 and 7). 1–2 repeats are often insufficient to obtain significant repression and 3–4 repeats may

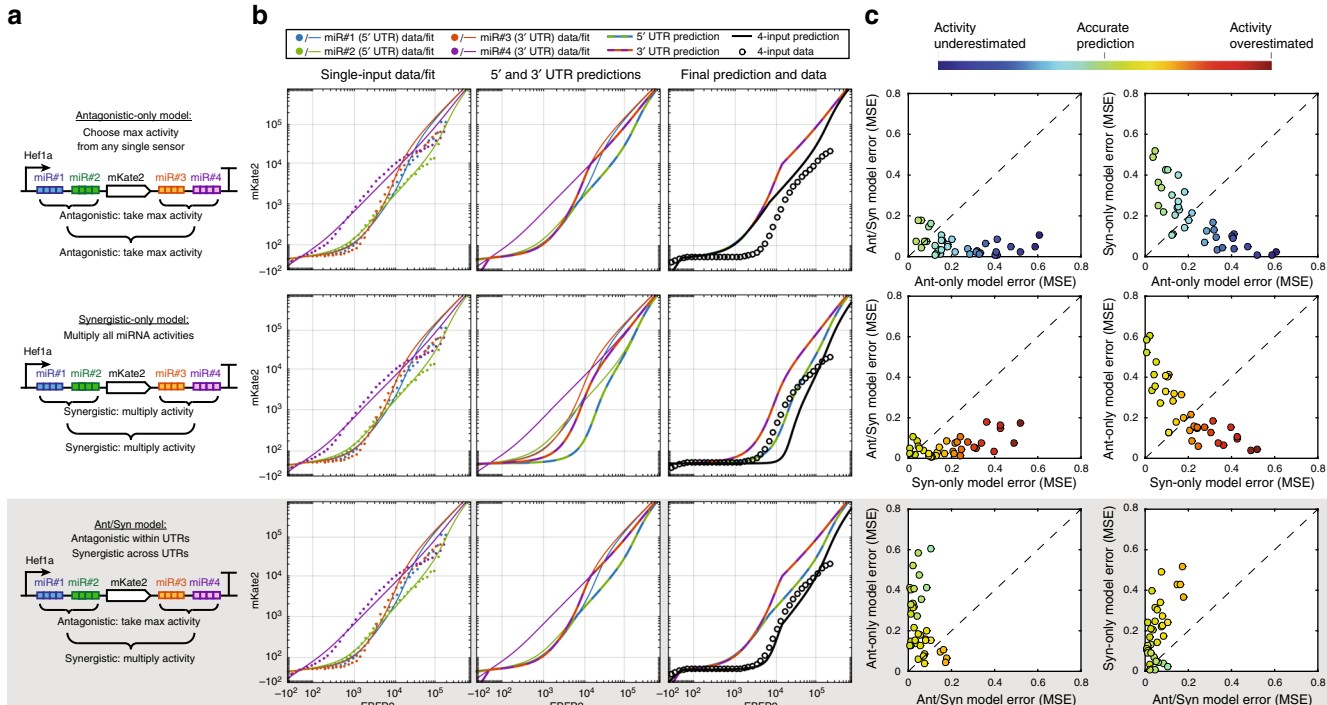

**Fig. 5** Ant/Syn model provides accurate predictions of 4-input miRNA classifiers. **a** Diagrams for 4-input sensors with associated logic for combining miRNA activity annotated for the antagonistic-only model (top row), synergistic-only model (middle row), and Ant/Syn model (bottom row). Two sets of distinct miRNA target sites were placed in each 5′ and 3′ UTR. **b** Example 4-input predictions based on single-input data. Rows across panels depict analysis of the sensor configuration in **a**. Single-input miRNA sensor data and model fits are shown for each of the four miRNAs (left column). Predictions are made separately for the 5′ and 3′ UTRs (center column) by taking the maximum activity for any single sensor (antagonistic-only model and Ant/Syn model) or alternatively by multiplying fold changes (synergistic-only model) in each transfection marker bin. Final predictions are made (right column) by combining activities from 5′ and 3′ UTRs by multiplying fold changes for each UTR (synergistic-only model and Ant/Syn model) or by taking maximum activity (antagonistic-only model). For this example, miR#1 = miR-17-5p, miR#2 = miR-31-3p, miR#3 = miR-519a-3p, and miR#4 = miR-16-5p. All 36 miRNA target combinations tested with associated predictions and data can be found in Supplementary Figs. 12–15. **c** Comparison of error in antagonistic-only, synergistic-only, and Ant/Syn model predictions for 36 different classifiers. Errors calculated as mean squared error between prediction and obtained data are plotted for each of the three possible comparisons. Additionally, points are colored by the degree of miRNA activity underestimation or overestimation according to the model plotted along the x axis (i.e., top row colored by antagonistic-only model errors, middle row by synergistic-only model error, bottom row by Ant/Syn model errors), resulting in six total graphs. For all 36 tested combinations of miRNAs, errors from the Ant/Syn model were similar or better than those from the other two models, as shown by most points falling within upper-left triangle. In comparison, the antagonistic-only model tended to underestimate repression and the synergistic-only model tended to overestimate repression

be required even with high activity miRNAs, suggesting possible roles of cooperativity on miRNA repression. Second, concatenating target 'sets' of different miRNAs within the 3′ UTR does not result in increased repression, indicating an antagonistic interaction where the combined activity reflects only the miRNA target set with the greatest activity. Third, target sets in both 5′ and 3′ UTRs exhibit synergistic interactions which can be approximated as the multiplication of fold repression contribution from the 3′ UTR with that from the 5′ UTR. Finally, antagonistic and synergistic interactions are combined into our proposed Ant/Syn model. Based on our observations, the Ant/Syn model is better able to explain data from a panel of 4-input classifiers than either the antagonistic-only or synergistic-only models. Our model represents a considerable advance compared to previous computational models of miRNA activity[22,23] in that it accurately models circuits that respond to multiple different miRNAs. We note that while our multi-input sensors and Ant/Syn model are most relevant for sensing applications, the data may also be useful for probing the limits of miRNA biology since our sensors are designed for maximal repression.

We examined the potential for cooperative effects when concatenating multiple target sites into sets[31]. Interestingly, we observed apparent cooperativity across target sites for miRNAs even when different target sites are interleaved to make new mixed target sets. In other words, a set of interleaved target sites for four different high activity miRNAs represses similarly to a set of four targets of the same high activity miRNA (Supplementary Fig. 7). This effect suggests that cooperativity could be mediated by Ago–Ago or Ago–protein–Ago interactions. One proposed protein mediator is GW182 which can interact with multiple Ago molecules simultaneously, forming a complex containing many Ago molecules[32–35]. When one miRNA target site-containing transcript is bound by Ago, other Ago molecules within the complex are more likely to bind the transcript, enabling cooperative repression (Supplementary Fig. 8). One potential complication for this model is that cleavage-independent repression mechanisms (e.g., mRNA deadenylation, target sequestration) should dominate over cleavage-dependent mechanisms. Otherwise each target site could be independently cleaved when bound by Ago2, leading to synergistic repression contributions for each target site. Since addition of exogenous siRNA along with complementary target sites was able to enhance repression of transcripts already containing three high activity target sets (i.e., siRNA targets act independently of miRNA targets), our data suggest that there could be two different repression mechanisms for siRNA and miRNA even when all target sites are perfectly

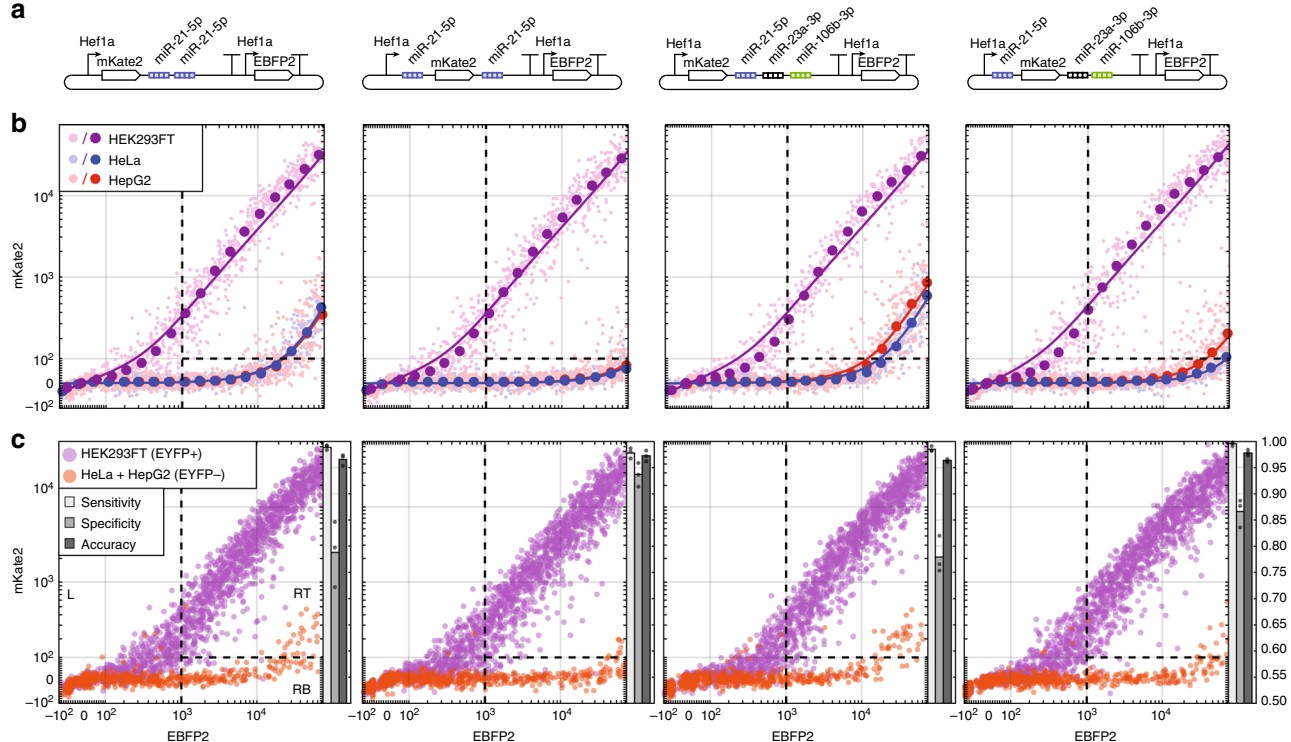

**Fig. 6** miRNA target site position affects cell classifier performance substantially. **a** Circuit diagrams for miRNA classifiers to distinguish HEK293FT cells from HeLa and HepG2. Two pairs of classifiers were tested, some encoding miRNA targets only in the 3′ UTR and others with one set of miR-21-5p targets moved to the 5′ UTR. miRNAs exhibiting high activity in HeLa and HepG2 cells but not HEK293FT were selected for inclusion. **b** Fluorescence observed after separate transfections of the corresponding classifiers from **a** into HEK293FT, HeLa, and HepG2 cells. Expression of mKate2 reporter remained high in HEK293FT cells for all classifiers, but knockdown of mKate2 in HeLa and HepG2 was enhanced only when miR-21-5p targets were placed in the 5′ UTR (columns 2 and 4) compared to the 3′ UTR (columns 1 and 3). Subsampled raw data are indicated as light points and binned data/fits are indicated as dark points/lines. Subsampling was performed to normalize the number of cells within each EBFP2 expression level bin in order to minimize effects of transfection efficiency on sensitivity and specificity measurements. **c** Classification of three cells lines in co-culture format. HEK293FT cells with genomically integrated constitutive expression of EYFP were co-cultured with HeLa and HepG2. Transfections were performed in cell mixtures for each classifier. After flow cytometry, gating for EYFP+ and EYFP− cells was used to determine whether each cell was HEK293FT or HeLa/HepG2 in origin. Dotted lines demarcate three regions used to determine whether cells were classified as HEK293FT (RT: right top), HeLa/HepG2 (RB: right bottom), or undetermined due to low transfection levels (L: left). Sensitivity or true positive rate is calculated using $[RT^+]/([RT^+] + [RB^+])$, specificity or true negative rate is calculated using $[RB^-]/([RB^-] + [RT^-])$, and accuracy is calculated using $([RT^+] + [RB^-])/([RT^+] + [RT^-] + [RB^+] + [RB^-])$ where + and − denote EYFP+ and EYFP−, respectively. Classifiers with separate target sites in 5′ and 3′ UTR were better able to distinguish different cell types by reducing off-target expression as indicated by greater specificity (>85%). Bar charts indicate means from three technical triplicates shown as black dots

complementary, with siRNA utilizing cleavage and miRNA using cleavage-independent mechanisms (Supplementary Fig. 9). While siRNA and miRNA are known to repress via different mechanisms for endogenous transcripts[36], mechanisms for synthetic sensors with perfectly complementary target sites have been less studied. Other relevant evidence in plant biology suggests that cleavage-independent mechanisms like translational inhibition are important even though most miRNA target sites in plants are perfectly complementary[37] and it has been suggested that non-cleaving repression may be the default mechanism for miRNA repression in both mammals and plants, despite near-perfect complementarity for the latter[38]. Additionally, examples of miRNA–RISC complexes lacking cleavage capability have been described previously[39].

In general we and others[40] have observed that miRNA target site sets placed in the 5′ UTR showed similar repression levels to those in the 3′ UTR. However, the role of 5′ UTR miRNA target sites in endogenous transcripts has not been studied extensively. Since there appears to be ample potential for effective miRNA regulation at the 5′ UTR, it appears likely that endogenous miRNA-mRNA targets within the 5′ UTR should be more abundant than the six validated examples currently described in the literature[41]. Several studies have shown miRNA repression in 5′ UTRs could be important in some transcripts, even though such repression is less-studied than that in the 3′ UTR despite hundreds of possible 5′ target sites revealed by computational predictions[6,40]. For comparison, the number of predicted target sites in the 3′ UTR is on the same order of magnitude (though several fold greater) when the same computational approach is used[6]. Analysis of CLIP-seq data has also identified hundreds of thousands of target sites in the CDS and 5′ UTR regions in mammals[42]. Thus, miRNA regulation of 5′ UTRs is an important but understudied aspect of miRNA biology.

Our observation of synergistic interactions across UTRs supports the notion that endogenous transcripts bearing miRNA targets in both UTRs, termed miBridges, would have enhanced repression[6]. Moreover, it may be possible that miBridges are not limited to transcripts bearing target sites for the same miRNA molecule in both UTRs, since we observed that different miRNA targets still exhibit synergistic interactions when placed in opposite UTRs (Figs. 4 and 5) and that combinations of different miRNAs act similarly to combinations with the same miRNA (Supplementary Fig. 7). In this case miBridges could be a more general and common motif in biology, where miRNA targets in

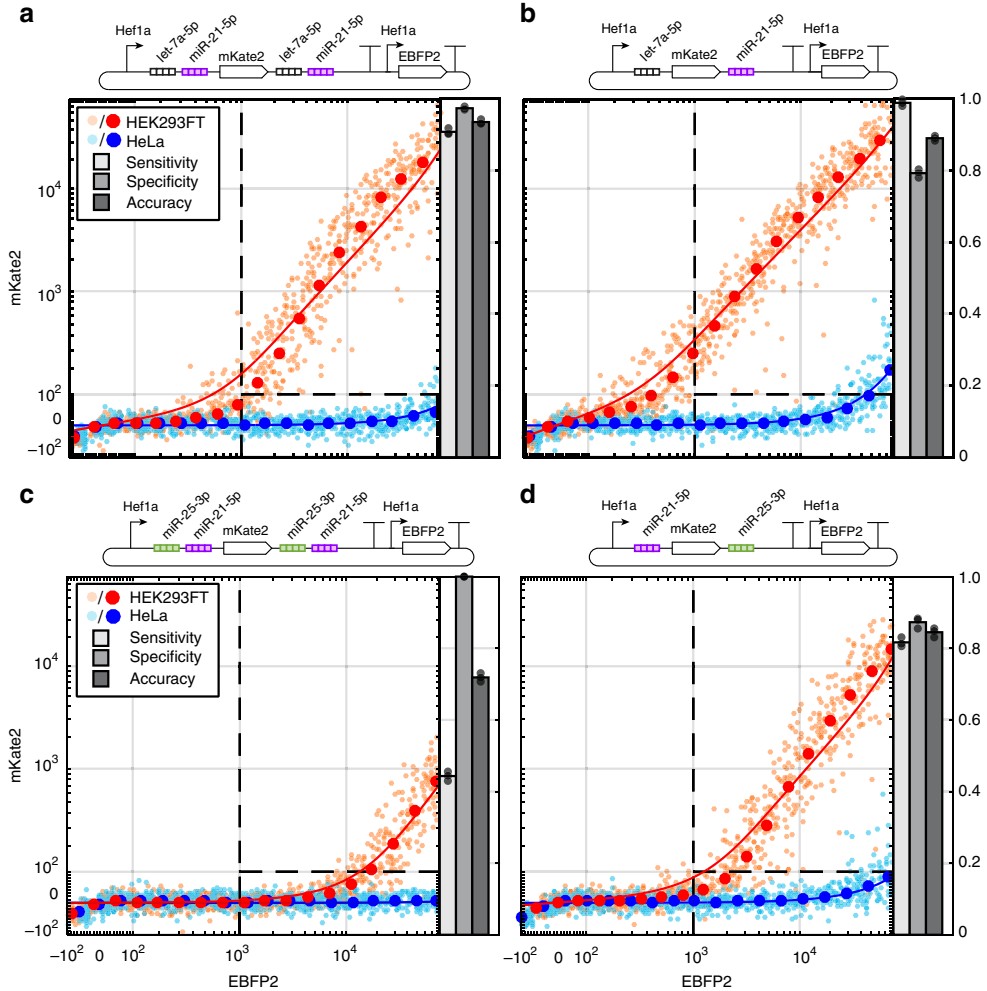

**Fig. 7** Ant/Syn model predicts the effects of target site position and number on the trade-off between sensitivity and specificity. Several classifiers were tested to further explore the effect of target set position and duplication on classifier sensitivity, specificity, and accuracy. We chose to sense three different miRNAs: miR-21-5p and let-7a-5p have high activity in HeLa and slight activity in HEK293FT, miR-25-3p has medium activity in HeLa and HEK293FT. In this example, miR-25-3p would be important for classifying against a third cell type not shown here (e.g., glioma[47]). **a**, **b** To test the effects of miRNA target set duplication on classification, versions of 2-input classifiers were built containing two target sets for let-7a-5p and miR-21-5p in both UTRs (**a**), or single sets in separate UTRs (**b**). Output in the classifier with four total target sets was dominated by synergy between miR-21-5p activity from both UTRs, resulting in very low output in HeLa and high specificity (97%) but with a slight trade-off in sensitivity (91%) since output in HEK293FT (where miR-21-5p activity is greater than let-7a-5p) was reduced with target site duplication. In contrast, high sensitivity (99%) but lower specificity (79%) was obtained with the classifier with only a single set of each miRNA target (**b**) since output in HEK293FT cells was attenuated by only one miR-21-5p target set. **c**, **d** Classifiers similar to **a** and **b** were built containing sets of target sites for miR-21-5p and miR-25-3p with duplication (**c**) and without (**d**). Again duplication resulted in high specificity (100%) but with a significant trade-off in sensitivity (44%) due to synergistic effects from both sets of miR-25-3p targets, which allows two medium activity target sets to exhibit high activity when combined in HEK293FT cells. Also, miR-25-3p targets contain an ATG sequence resulting in further reduction in output. In contrast, the classifier with only two target sets avoided multiplicative effects from miR-25-3p and introduction of uORFs. These effects result in greatly improved sensitivity (82%) at a cost of slightly reduced specificity (87%). The latter classifier also showed higher overall accuracy (84% compared to 72%)

the 5′ UTR enhance regulation in the 3′ UTR, similarly to measured effects from targets in the CDS to those in the 3′ UTR[5].

There are several possible explanations for the mechanistic basis for the Ant/Syn behavior we observed. We tested whether steric effects (Supplementary Fig. 2), inherent biological repression limits (Supplementary Fig. 9), and miRNA sponging (Supplementary Fig. 10) could explain the antagonistic interactions, though none were able to. The observation of distinct synergistic and antagonistic behavior based on whether target sites are on separate or the same UTRs suggests that distinct mechanisms for miRNA repression may exist for targets in the 5′ UTR compared to the 3′ UTR, otherwise we would expect to observe only

antagonistic effects even with target sets in separate UTRs. The repression effects we observed could be explained if target sites in separate UTRs separately saturate distinct repression machinery (e.g., different subunits of the CCR4-NOT complex mediating deadenylation or translational repression). Indeed, numerical simulation of models incorporating shared or separate repression machinery were able to recapitulate the antagonistic and synergistic interactions, respectively (Supplementary Fig. 11). Since targets in the 3′ UTR are generally considered to mediate mostly mRNA destabilization[43], it is possible that targets in the 5′ UTR act mostly to translationally repress the targeted transcript (Supplementary Fig. 8). Due to the proximity of 5′ UTR targets to translational machinery, it may be possible that a majority of

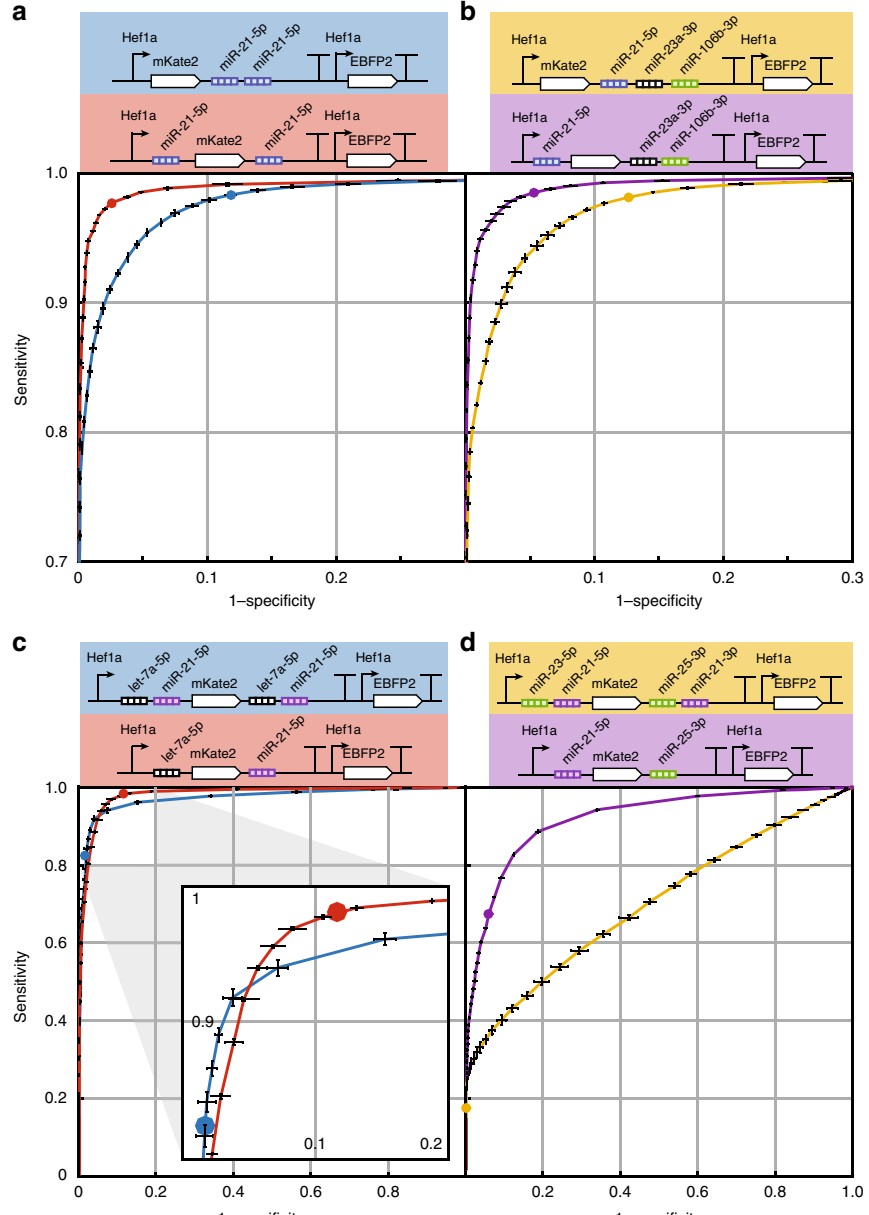

**Fig. 8** ROC curves for classifiers illustrate specificity and sensitivity tuning by miRNA target set location and number. Receiver operator characteristic (ROC) curves derived from data in Figs. 6 and 7 were generated by calculating sensitivity and specificity at varying the mKate2 thresholds (previously fixed at 10$^2$). For reference, circles along the curves show sensitivity and specificity at the fixed threshold used for Figs. 6 and 7. **a** Circuit diagrams and ROC curves for miRNA classifiers with varying target site position as tested in Fig. 6. Circuit shading color corresponds to ROC curve color. The classifier with miR-21-5p target sets in different UTRs (red) shows greater maximal accuracy compared to that with target sites in only the 3′ UTR (blue) as indicated by nearer approach towards the upper left and greater area under the curve (AUC = 0.9947 vs 0.9895). **b** Similarly, the 3-input classifier with miR-21-5p in the 5′ UTR and miR-23a-3p and miR-106b-3p in the 3′ UTR (purple) exhibits better overall classification than the variant with target sets in only the 3′ UTR (yellow), with AUCs of 0.9952 and 0.9881, respectively. **c** Circuit diagrams and ROC curves for miRNA classifiers with or without target site duplication across UTRs for let-7a-5p and miR-21-5p. Data for **c** and **d** correspond to that from Fig. 7. Both classifier variants show similar maximum accuracy when classifying between HEK293FT and HeLa cells, though at the example threshold of 10$^2$ the variant without duplication (red, AUC = 0.9833) prioritizes sensitivity while the variant with target set duplication (blue, AUC = 0.9750) prioritizes specificity (see inset for magnified view). **d** For classifiers with target sites for miR-25-3p and miR-21-5p, the circuit without duplication (purple, AUC = 0.9235) exhibits better classification compared to that with duplication (yellow, AUC = 0.7018), since repression was determined by both miR-25-3p and miR-21-5p rather than being dominated by miR-21-5p and also since potential uORFs are avoided by removing miR-25-3p targets from the 5′ UTR (both effects enhance sensitivity). Error bars indicated s.d. calculated by threshold averaging from three technical triplicates

repression there is due to translational repression. Interestingly, for the RNA binding protein LIN41 in *C. elegans*, different modes of repression have been observed depending on UTR location (i.e., translational repression in the 5′ UTR and mRNA destabilization in the 3′ UTR)[44]. Results for LIN41 combined

with our data, provide the attractive hypothesis that other RNA binding proteins like Ago may use a similar mechanism to mediate different modes of repression depending on location within the UTRs. Our results highlight the need for further studies on elucidating differences between repression from the

5′ UTR, 3′ UTR, and coding region for both synthetic miRNA sensors and endogenous transcripts[1].

In summary, we describe the development of a workflow that uses an Ant/Syn model, which takes into account miRNA target set number, location, and interactions, in order to accurately predict the behavior of multi-input miRNA sensors from well-characterized single-input sensors. In contrast to previous methods yielding fold enrichment metrics[26], our characterization provides highly detailed information of miRNA activity across a wide spectrum of transduction efficiencies and output levels, allowing accurate predictions of multi-input miRNA classifier function using our Ant/Syn miRNA repression model. This model contributes to synthetic biology by enabling predictable responses to miRNA inputs, and to biology by showing that miRNA repression must take into account location and number of target sites.

## Methods

**Construction of miRNA sensor library**. A common low sensor backbone (LSB) plasmid was constructed using a Gateway-Gibson strategy[45] to enable assembly of miRNA low sensor libraries in a single Golden Gate[27] step. Long DNA oligonucleotides encoding miRNA target site repeats were synthesized as ultramers by IDT, annealed and restriction-ligated via BbsI-mediated Golden Gate. White/blue screening enabled selection of mostly correct clones which were further verified by Sanger sequencing. Relevant plasmid and oligonucleotide sequences are given in Supplementary Datas 1–3. Overhangs for Golden Gate steps are listed in Supplementary Table 4.

**Construction of miRNA classifiers**. Several plasmid backbones were constructed to expedite assembly of miRNA classifiers. JG107 and JG108 plasmids were assembled similarly to LSB and contain a LacZ cassette into which miRNA target sites were cloned using BbsI-mediated Golden Gate. An important feature for these plasmids is that their miRNA target sites possess distinct overhangs when digested with *Bsa*I. The low sensor backbone reverse (LSBr) plasmid retains the same design of LSB but with *Bsa*I and *Bbs*I recognition sites reversed. The LSBr-5 and 3UTR construct introduces a further addition of a *Bsa*I/*Bbs*I-flanked mCherry selection cassette in the 5′ UTR of the mKate2 transcription unit. miRNA target sites present in LSB, JG108, and JG107 were then used as inputs to clone 3-input classifiers into LSBr using BsaI-mediated Golden Gate. 4-input classifiers were cloned into LSBr-5and3UTR using a similar strategy, with target sites for the 3′ UTR originating from LSB and JG108 plasmids while JG107 encoded an inert sequence. Target sites were cloned into the 5′ UTR from annealed oligonucleotides flanked by *Bsa*I sites. White colonies were selected and verified by Sanger sequencing

**Cell culture**. HEK293FT were purchased from Thermo Fisher, HeLa and HepG2 cells were obtained from ATCC. HEK293FT and HeLa cells were grown in DMEM with 4.5 g/L glucose, L-glutamine, and sodium pyruvate (Cellgro) supplemented with 10% characterized FBS (HyClone). HepG2 cells were grown in DMEM with high glucose, 2 mM L-glutamine, without sodium pyruvate (HyClone) supplemented with 10% characterized FBS (HyClone). All cell lines were grown at 37 °C and 5% $CO_2$. All cell lines tested negative for mycoplasma.

**Transfection of miRNA sensors and classifiers**. miRNA low sensor libraries were transfected into cells using a reverse transfection method[28]. Example volumes are provided for a 100 μL mix which was sufficient for roughly 25 wells in 96-well plate format. 2 or 0.2% gelatin (Sigma-Aldrich) was diluted to 0.05% in sterile water. 1 volume of 2 M sucrose was added to 9 volumes of buffer EC from an Effectene transfection kit (Qiagen) to make a master mix with total volumes depending on how much DNA was transfected. A total of 37.5 μL Buffer EC mixture was added to 2 μg of DNA sample for a final DNA concentration of 20 ng/μL after addition of all components (DNA, buffer EC, enhancer, Effectene, gelatin). Overall, 3.75 μL of enhancer solution was added to the DNA/EC mixture, mixed gently, and incubated at room temperature for 5 min. A total of 12.5 μL of Effectene was added and mixed followed by incubation for 10 min at room temperature. 53.75 μL of 0.05% gelatin was added and mixed gently to obtain the final transfection mix. A total of 4 μL of the mixture was added to each well in a 96-well plate, wells were covered with sterile adhesive aluminum covers, and stored at -80 °C without dessication. Cells were reverse transfected by dissociating and counting cells as normal, followed by seeding of cells at usual concentration for forward transfection in 100 μL of cell media.

miRNA classifiers were transfected using forward transfection methods with lipofectamine 3000 (Thermo Fisher) according to the manufacturer's protocols. All transfections were conducted with 400 ng of DNA, 1 μL of P3000 reagent, 1 μL of lipofectamine 3000 in Opti-MEM (Thermo Fisher). HEK293FT cells were seeded in 24-well plates at various densities in 0.5 mL volume of media (HEK293FT = 2 ×

$10^5$ cells/well, HeLa = $1 \times 10^5$ cells/well, HepG2 = $1.5 \times 10^5$ cells/well). For cell mixtures, $5 \times 10^4$ of each cell type was seeded per well and a 1:3 ratio of DMEM high glucose:DMEM was used. miR-21-5p mimic was obtained from Sigma-Aldrich and cotransfected using the same method as for DNA. Target site combinations for 4-input sensors are listed in Supplementary Table 2.

**Flow cytometry**. Flow cytometry was conducted using an LSR Fortessa cytometer with 405, 488, and 561 nm lasers (BD Biosciences). Reverse transfected cells in 96-well plate format were analyzed using the high throughput sampler option to collect >10,000 cells as quickly possible. Forward transfected cells in 24-well plate format were analyzed in tube format with >50,000 events collected per sample. mKate2 was detected using a 561 nm laser with 610/20 filter and 235 PMT voltage, EBFP2 was detected using a 405 nm laser with 450/50 nm filter and 240 PMT voltage, and EYFP was detected using a 488 nm laser with 530/30 nm filter and 200 PMT voltage.

**Model-based fitting and prediction of miRNA activities**. Models and predictions were implemented in MATLAB with steady-state assumptions. Briefly, cytometry files for single-input sensors were read in, gated using forward and side scatter, binned by EBFP2 fluorescence, and parameters fit to the model using lsqcurvefit. Parameter fits were used to generate predictions either by multiplication of fold repression (synergistic-only model) or taking the minimization of mKate2 expression (antagonistic-only model) or a combination of the two (Ant/Syn model). MATLAB scripts for analysis are provided in Supplementary Data 4 and rate constants used are listed in Supplementary Table 3.

**Statistical analysis**. Statistics are based on a log-normal distribution in fluorescence with a geometric standard deviation of approximately 2.8, an assumption supported by all samples in the flow cytometry data we collected. Greater than 50,000 events were chosen to allow at least 100 data points per fluorescent bin, and binned data are presented as medians. Unless otherwise indicated, bar plots indicate means from the three technical triplicates overlayed on the same graph. For ROC curves, error bars were calculated using threshold averaging and again indicate standard deviations from technical triplicates.

**Code availability**. MATLAB scripts/functions are provided as Supplementary Data 4 and are also available from Github at the following link https://github.com/jeremygam/multi-input_miRNA_sensor_prediction.

**Data availability**. All data files supporting this study are available on request from the corresponding author. DNA sequences are provided in Supplementary Datas 1–3 and GenBank accession number MH210859 while physical plasmids will be available from Addgene (#103144-103770).

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

## Acknowledgements

We would like to acknowledge Douglas Lauffenburger, Philip Sharp, Peter Carr, Cameron McBride, Ross Jones, Breanna DiAndreth, Jin Huh, Sophie Strobel, David Kong, Todd Thorsen, David Walsh, and Scott Wick for discussion; Breanna DiAndreth for MATLAB scripts and proofreading; Zhen Xie, Yinqing Li, Xavier Duportet, and Patrick Guye for plasmids and materials; Aishwarya Jagtap for technical assistance; and Olga Parkin, Stephen Firsing, John Scarpa, and Cameron Haase-Pettingell for administrative support. This work was supported by grants from the NIH (R01CA173712 and P50GM098792) and the NSF (award number 1745645), and partially by Cancer Center Support (core) Grant P30CCA14051 from the NCI.

## Author contributions

J.J.G. conceived the ideas, designed and performed experiments, analyzed data, and wrote the manuscript. J.B. and R.W. conceived the ideas and wrote the manuscript.

## Additional information

**Competing interests:** The authors declare no competing interests.

