## [Peer Review File · Nature Communications]

Reviewers' comments:

Reviewer #1 (Remarks to the Author):

The authors describe an integrated experimental and modelling approach to predict the effect of miRNA binding sites on reporters with the aim of building improved sensors based on the miRNA composition of the cell. The authors built a miRNA sensor library comprising about 600 plasmids, in addition to a set of miRNA classifiers with different distributions of miRNA binding sites across 3' and 5' UTR. The authors then used measurements on single-miRNA sensors to predict the function of multi-input miRNA sensors. To this end the authors proposed a novel antagonistic/synergistic model and validated it experimentally to build a miRNA classifier for HeLa and HepG2 cells. This manuscript will be of interest to the synthetic biology community, potentially it would be of interest also to molecular biologists but the speculations on the different mode of action of miRNA on the 3' and 5' UTR are not substantiated making the manuscript of interest to a more restricted readership.

Major points

1) The authors mention a workflow they named miCAD in the introduction and the discussion but no description of it can be found nor in the main text and neither in supplementary material. Moreover no link to a software repository or online website is given. It is not clear to me what miCAD is and how it relates to the Ant/Syn model. The authors have to explain better what miCAD is or remove any reference to it. For example, is miCAD a set of scripts to analyse the experimental data? if so this is not of general interest hence I would remove any mention of it, or is miCAD a self-contained software to predict sensor activity from a plasmid design using the Ant/Syn model? if so the authors should provide a usable workflow via a repository (Github) or a Docker or an online tool implementing their workflow.

2) The reproducibility of the single miRNA sensor library in Fig S1E reports the M/Km metric over two experimental replicates in HEK cells. The authors should report the correlation coefficient in the figure and also its pvalue. It seems that reproducibility is acceptable only for M/Km values greater than 0.1 indicating that only estimated activities above this level are reproducible.

Minor points

a) Introduction and Discussion could be shortened.

b) typos, e.g. page 4 third line from last "ASE expected..."

Reviewer #2 (Remarks to the Author):

The authors attempt to decipher the combinatorial miRNA code and their context-dependent interactions. I think this is a very strong and well written paper. The results are well-justified and clear. I think this paper should be published as is.

Reviewer #3 (Remarks to the Author):

This is a well-written paper describing a phenomenological model of the interaction between cellular miRNA and synthetic target constructs. The model is validate using a wide range of reporters with numbers and combinations of targets sites inserted both in the 5' and 3'UTR. Provided the authors address the comments below, the paper should be published in Nature Communications.

* In the first sentence of the Results the authors state "we designed, constructed and tested a miRNA sensor library..." The authors should state more clearly how many of these sensors were functionally validated (i.e. repression was measured with specific miRNA inputs). Presumably only a subset of the 620 miRNA for which sensors were built are actually expressed in the cell lines used in this study.

* It would be good to add a more detailed and intuitive description of the different models (antagonistic/additive/synergistic) to the main text. Readers may not be familiar with the Chou-Talalay method. Maybe also rewrite Fig. 2b to use $n=2$ and explicitly write out all terms while moving the more general expression to the supplement.

* The authors should more specifically address the range of applicability and possible limitations of their model in the context of miRNA biology. Key sensor design choices do not reflect known biology (although these choices are optimal for sensing applications):

**The Ant/Syn predictions seem to hold for combinations of "sets" of miRNA target sites rather than individual target sites. As the authors write in the very beginning of the discussion "concatenation of miRNA target sites in a UTR increases repression of a target reporter up to approximately 4 repeats,"

a result more consistent with an additive rather than antagonistic model. In fact, such approximate additivity between multiple miRNA sites has been observed for several human 3'UTRs. It might be advantageous to expand to the phenomenological model such that it captures both regimes. (Although it is pointed out that many mRNA 3'UTRs contain large numbers (>4) of target sites, it is not clear whether such UTRs would be described by the ANT/SYN model since only a subset of these sites will have matching miRNA in most cell types).

**As the authors know, most human miRNA bind their target only by a short seed sequence. Although using multiple fully complementary targets is the right choice for sensing it also means that the measurements and resulting model are less directly applicable to miRNA biology.

* Although not required, it would be very interesting to combine a fluorescence-based reporter assay with mRNA-seq to understand to what degree a reduction in fluorescence due to miRNA targeting is correlated with a reduction in the mRNA levels. In particular it would be interesting to see whether targets in the 5' and 3'UTRs have the same effect on mRNA stability.

Point-by-point Response

Reviewer #1:

The authors describe an integrated experimental and modelling approach to predict the effect of miRNA binding sites on reporters with the aim of building improved sensors based on the miRNA composition of the cell. The authors built a miRNA sensor library comprising about 600 plasmids, in addition to a set of miRNA classifiers with different distributions of miRNA binding sites across 3' and 5' UTR. The authors then used measurements on single-miRNA sensors to predict the function of multi-input miRNA sensors. To this end the authors proposed a novel antagonistic/synergistic model and validated it experimentally to build a miRNA classifier for HeLa and HepG2 cells. This manuscript will be of interest to the synthetic biology community, potentially it would be of interest also to molecular biologists but the speculations on the different mode of action of miRNA on the 3' and 5' UTR are not substantiated making the manuscript of interest to a more restricted readership.

Response: We believe that the Ant/Syn model serves as a one of many steps towards more comprehensive model and the insights gained from this study on the potential different modes of action of miRNA is of interest. Future steps for synthetic and molecular biologists working with miRNA will be to examine the effect of miRNA target position and structure on repression mechanism. Our work aims in part to bring together previous biological studies on mechanism (e.g. Lytle 2007, Lee 2009) with quantitative predictive models, in order to better understand relative contributions of each mechanism to miRNA regulation. We have made several small alterations throughout the narrative and have included the following text early on in the Introduction in order to

clarify this goal: “We believe that the modeling approaches in this study, combined with other future studies on how miRNA repression mechanism is modulated by miRNA target position and sequence, will be essential for deeper understanding of the intricacies of miRNA regulation.” **We have also modified the Abstract to include the following:** “These results contribute to, and motivate research on, the aforementioned understudied but important aspects of miRNA biology.”

Comment 1: The authors mention a workflow they named miCAD in the introduction and the discussion but no description of it can be found nor in the main text and neither in supplementary material. Moreover no link to a software repository or online website is given. It is not clear to me what miCAD is and how it relates to the Ant/Syn model. The authors have to explain better what miCAD is or remove any reference to it. For example, is miCAD a set of scripts to analyze the experimental data? if so this is not of general interest hence I would remove any mention of it, or is miCAD a self-contained software to predict sensor activity from a plasmid design using the Ant/Syn model? if so the authors should provide a usable workflow via a repository (Github) or a Docker or an online tool implementing their workflow.

Response: We appreciate the reviewer’s suggestions on clarifying the description of our workflow. In order to reduce confusion, we have removed mentions of the term miCAD and instead refer to more general and readily understandable steps/components in the workflow like “model” and “analysis scripts”. We have made sure that analysis scripts are readily accessible to those interested via Github and in a supplementary zip file. Information is listed in the Data Availability section as follows: “DNA sequences and MATLAB scripts/functions are provided as supplementary files while plasmids will be available from Addgene. Analysis and predictions scripts are available from Github at the following link https://github.com/jeremygam/multi-input_miRNA_sensor_prediction”

Comment 2: The reproducibility of the single miRNA sensor library in Fig S1E reports the M/Km metric over two experimental replicates in HEK cells. The authors should report the correlation coefficient in the figure and also its pvalue. It seems that reproducibility is acceptable only for M/Km values greater than 0.1 indicating that only estimated activities above this level are reproducible.

Response: We thank the reviewer for the helpful technical feedback. We have calculated the correlation coefficients and p-values for replicated sensor measurements and compared reproducibility for either low or high activity sensors. We clarify that reproducibility is reduced at low activity, likely due to noise in model fits as sensor performance in low ranges (0.01-0.1 M/Km) is substantially the same, with mKate/EBFP2 curves showing almost no repression. We have added the following information and panel F to Supplementary Figure 1 to clarify points raised by the reviewer: “Replicates showed a correlation coefficient of $R=0.85$ and significantly small p-value, rejecting the null hypothesis that sensor data from replicates was uncorrelated (calculated on log transformed data; linear data also show significant correlation of $R=0.82$, $p=4.5e-95$). (F) Correlation coefficient and p-values were calculated for groups of either high activity ($M/Km > 0.1$, red) and low activity ($M/Km < 0.1$, blue) sensors in HEK293FT. Both groups show statistically significant correlation though there was a large reduction in correlation for low activity sensors, limiting reproducibility in low activity ranges. However, it should be noted that miRNA activities within this low regime (M/Km between 0.01 and 0.1) behave similarly with little detectable repression. Variance for low activity measurements may be due to noise in model fitting or amplified by the log transform.”

Minor Comment A: Introduction and Discussion could be shortened.

Response: We have removed redundancy in the discussion of the possible mechanism of antagonistic and synergistic interactions. We have also moved less important sections to a Supplementary Discussion section so that more essential information is included within the introduction and discussion. The following passages were moved to the supplementary:

“miRNA repression mechanisms are varied and complex, with molecular players including the Argonaut proteins, GW182, PABPC, the CCR-NOT complex, the PAN2-PAN3 complex, and decapping proteins.\cite{Jonas2015} Recent evidence shows even more possible interactions with DDX6, phosphorylation by CSNK1A1 and dephosphorylation by the ANKRD52-PPP6C complex.\cite{Golden2017,Rouya2014} Further complicating matters is the fact that miRNAs can mediate multiple modes of regulation including deadenylation, decapping, cleavage, and translational repression.

Our results support previous preliminary findings the repression mechanisms for targets in the 5' UTR and 3' UTR may be distinct (e.g. translational repression in the 5' UTR and mRNA destabilization in the 3' UTR).

Also, we hypothesize that antagonistic interactions could act to reduce noise and increase redundancy in highly regulated genes, since if the miRNAs have similar activity, a decrease in activity of any single miRNA would not have significant impact on gene expression. This is in contrast to the stereotypical additive or independent (i.e. synergistic) models where fluctuation of any single miRNA would have effects on gene expression, propagation of noisy miRNA activity toward target gene expression. This mechanism of noise reduction likely works in tandem with other modes where miRNAs can suppress noise at the network level.\cite{Siciliano2013} As such, it may be possible that highly miRNA-regulated genes use antagonistic interactions within the same UTR to help reject noise and use synergistic interactions across UTRs to increase repression.”

Minor Comment B: typos, e.g. page 4 third line from last "ASE expected..."

Response: We have found and fixed several typos; we appreciate the reviewer pointing them out.

Reviewer #2:

The authors attempt to decipher the combinatorial miRNA code and their context-dependent interactions. I think this is a very strong and well written paper. The results are well-justified and clear. I think this paper should be published as is.

Response: We would like to thank the reviewer for taking the time to review our manuscript and appreciate the feedback.

Reviewer #3:

This is a well-written paper describing a phenomenological model of the interaction between cellular miRNA and synthetic target constructs. The model is validate using a wide range of reporters with numbers and combinations of targets sites inserted both in the 5' and 3'UTR. Provided the authors address the comments below, the paper should be published in Nature Communications.

Comment 1: In the first sentence of the Results the authors state "we designed, constructed and tested a miRNA sensor library..." The authors should state more clearly how many of these sensors were functionally validated (i.e. repression was measured with specific miRNA inputs). Presumably only a subset of the 620 miRNA for which sensors were built are actually expressed in the cell lines used in this study.

Response: We thank the reviewer for pointing this out. In the initial sentence we now instead indicate that we sequence validated the sensor library, and then in the fourth paragraph we provide information about how many sensors exhibited activity in the tested cell lines. "Most miRNA sensors exhibited low or no miRNA activity (396 of 620 sensors or 64%), consistent with previous reports of miRNA sensor measurements in several cell lines (Supplementary Fig. S1). [Mullokanov2012]"

Comment 2: It would be good to add a more detailed and intuitive description of the different models (antagonistic/additive/synergistic) to the main text. Readers may not be familiar with the Chou-Talalay method. Maybe also rewrite Fig. 2B to use $n=2$ and explicitly write out all terms while moving the more general expression to the supplement.

Response: We appreciate the feedback on how to make the explanation of the Chou-Talalay method more accessible to readers. We have implemented the suggested changes to Figure 2 to make equations more easy to understand, while moving and adding details for the more general equations to the Modeling section in the supplementary. We have also included the following material in the Results section to explain synergistic, additive, and antagonistic interactions more intuitively with an example: "Additive predictions were made assuming mutually exclusive inhibitors, antagonistic predictions were made using similar assumptions but with a Hill coefficient value near zero, and synergistic predictions were made by multiplying contributions from each target site set (Fig. 2b,c). Here additive interactions indicate that combined repression from multiple inhibitors is equal to the 'sum' of their separate effects. For example, with two miRNAs giving 50% knockdown at a given plasmid concentration, additive interaction would give a final knockdown of 67%. Note that final knockdown is not 75% since that would indicate independent contributions to knockdown, which is a synergistic interaction. By definition synergistic and antagonistic interactions mark a deviation from the additive case. For the Ant/Syn model and taking the previous example of two miRNAs with 50% knockdown, independent (i.e. synergistic) contributions result in the intuitive 75% knockdown, while for the antagonistic case knockdown would be 50% since only the highest activity miRNA contributes to repression"

Comment 3: The authors should more specifically address the range of applicability and possible limitations of their model in the context of miRNA biology. Key sensor design choices do not reflect known biology (although these choices are optimal for sensing applications):

The Ant/Syn predictions seem to hold for combinations of "sets" of miRNA target sites rather than individual target sites. As the authors write in the very beginning of the discussion "concatenation of miRNA target sites in a UTR increases repression of a target reporter up to approximately 4 repeats," a result more consistent with an additive rather than antagonistic model. In fact, such approximate additivity between multiple miRNA sites has been observed for several human 3'UTRs. It might be advantageous to expand to the phenomenological model such that it captures both regimes. (Although it is pointed out that many mRNA 3'UTRs contain large numbers (>4) of target sites, it is not clear whether such UTRs would be described by the ANT/SYN model since only a subset of these sites will have matching miRNA in most cell types).

As the authors know, most human miRNA bind their target only by a short seed sequence. Although using multiple fully complementary targets is the right choice for sensing it also means that the measurements and resulting model are less directly applicable to miRNA biology.

Response: We thank the reviewer for the comments and agree that a model that accurately predicts effects from both regimes of >4 and <4 target sites would be very useful. Here we focus on highly repressed constructs for two reasons: their relevance in sensing applications, and the ability to gain information about the biological limits of repression, which has not been explored in as much depth compared to the more traditional repression from seed matches. We are currently working towards extending our model towards prediction of 1-4x perfect target sites, though since these effects can be nonlinear (Supplementary Figure S2) additional parameters will be needed to increase accuracy in that regime. We anticipate that many more measurements of sensors with 1-4x target sites will be required before we can determine how many additional parameters would be necessary, whether they would be general across different miRNA species or more miRNA-specific, and what their values would be. Ideal future models would be able to make repression predictions for synthetic or endogenous UTRs, given measurements of miRNA concentration or activity. Our current model is a significant step forward for designing and predicting synthetic UTRs, and should hopefully (along with future studies) inform some higher order rules about miRNA regulation that apply to the endogenous UTRs. We have included a discussion of this point within the Supplementary Discussion: “An ideal future model would be able to take a given UTR sequence (either endogenous or a synthetic design) and predict the extent of repression expected from miRNA regulation. Such a model would need to be able to find miRNA target sites computationally, determine the degree of binding and repression mode based on sequence complementarity, and also predict synergistic, additive, or antagonistic effects from multiple target site interactions. Other requirements would be measurements of miRNA concentration and activity in cell lines of interest. Our work here serves as a step forward for understanding target site interactions in the simplest and highest repressing synthetic systems, in addition to contributing data about miRNA concentration and activity. We anticipate that future work can build upon the Ant/Syn model to include more diverse types of sequences (different target site architectures, UTRs that contain between 1-4 target sites), in order to eventually be able to predict miRNA repression of any UTR.” We hope that this Supplementary Discussion section clarifies to readers how our sensors and models are designed for sensing applications, but will also serve as a starting point for us and others to perform future experiments that directly probe endogenous miRNA biology based on the principles uncovered by our study.

We have also included the following text early in the Discussion to clarify the relevance of the model toward sensing applications and miRNA biology: “We note that while our multi-input sensors and Ant/Syn model are most relevant for sensing applications, the data may also be useful for probing the limits of miRNA biology since our sensors are designed for maximal repression.”

Comment 4: Although not required, it would be very interesting to combine a fluorescence-based reporter assay with mRNA-seq to understand to what degree a reduction in fluorescence due to miRNA targeting is correlated with a reduction in the mRNA levels. In particular it would be interesting to see whether targets in the 5' and 3'UTRs have the same effect on mRNA stability.

Response: We thank the reviewer for this suggested experiment and agree that mRNA-seq results could further validate our reporter assays and provide additional information on mRNA stability. We plan to investigate similar experiments to refine our model in a following study (to be published separately) in order to enable increasingly sophisticated genetic circuit design approaches and to increase the general biological applicability of our model.

We would like to thank the reviewers one more time for their valuable feedback. We believe that we have addressed each of the comments from the reviewers and that the resulting changes have significantly enhanced the readability and coherence of our work. With the addition of new and clarifying text in the main and supplementary, we believe our manuscript is now appropriate for publication in *Nature Communications*.

Thank you again for your consideration of our manuscript.

Sincerely,

Ron Weiss
Director, Synthetic Biology Center
Professor
Department of Biological Engineering
Department of Electrical Engineering and Computer Science
Massachusetts Institute of Technology

REVIEWERS' COMMENTS:

Reviewer #3 (Remarks to the Author):

The authors have addressed my concerns and the paper can be published as is.